# A Fatal Case of *Pseudomonas aeruginosa* Community-Acquired Pneumonia in an Immunocompetent Patient: Clinical and Molecular Characterization and Literature Review

**DOI:** 10.3390/microorganisms11051112

**Published:** 2023-04-24

**Authors:** Nicole Barp, Matteo Marcacci, Emanuela Biagioni, Lucia Serio, Stefano Busani, Paolo Ventura, Erica Franceschini, Gabriella Orlando, Claudia Venturelli, Ilaria Menozzi, Martina Tambassi, Erika Scaltriti, Stefano Pongolini, Mario Sarti, Antonello Pietrangelo, Massimo Girardis, Cristina Mussini, Marianna Meschiari

**Affiliations:** 1Infectious Diseases, Azienda Ospedaliera-Universitaria of Modena, University of Modena and Reggio Emilia, 41125 Modena, Italy; nicole.barp94@gmail.com (N.B.);; 2Internal Medicine, Azienda Ospedaliera-Universitaria of Modena, University of Modena and Reggio Emilia, 41125 Modena, Italy; 3Intensive Care Unit, Azienda Ospedaliera-Universitaria of Modena, University of Modena and Reggio Emilia, 41125 Modena, Italy; 4Microbiology, Azienda Ospedaliera-Universitaria of Modena, University of Modena and Reggio Emilia, 41125 Modena, Italy; 5Risk Analysis and Genomic Epidemiology Unit, Experimental Zooprophylactic Institute of Lombardy and Emilia-Romagna, 43126 Parma, Italy

**Keywords:** community-acquired pneumonia, *Pseudomonas aeruginosa*, antibiotic therapy, whole genome sequencing

## Abstract

Rare cases of *Pseudomonas aeruginosa* community-acquired pneumonia (PA-CAP) were reported in non-immunocompromised patients. We describe a case of *Pseudomonas aeruginosa* (PA) necrotizing cavitary CAP with a fatal outcome in a 53-year-old man previously infected with SARS-CoV-2, who was admitted for dyspnea, fever, cough, hemoptysis, acute respiratory failure and a right upper lobe opacification. Six hours after admission, despite effective antibiotic therapy, he experienced multi-organ failure and died. Autopsy confirmed necrotizing pneumonia with alveolar hemorrhage. Blood and bronchoalveolar lavage cultures were positive for PA serotype O:9 belonging to ST1184. The strain shares the same virulence factor profile with reference genome PA01. With the aim to better investigate the clinical and molecular characteristics of PA-CAP, we considered the literature of the last 13 years concerning this topic. The prevalence of hospitalized PA-CAP is about 4% and has a mortality rate of 33–66%. Smoking, alcohol abuse and contaminated fluid exposure were the recognized risk factors; most cases presented the same symptoms described above and needed intensive care. Co-infection of PA-influenza A is described, which is possibly caused by influenza-inducing respiratory epithelial cell dysfunction: the same pathophysiological mechanism could be assumed with SARS-CoV-2 infection. Considering the high rate of fatal outcomes, additional studies are needed to identify sources of infections and new risk factors, along with genetic and immunological features. Current CAP guidelines should be revised in light of these results.

## 1. Introduction

In the most recent American Thoracic Society and the Infectious Diseases Society of America (ATS/IDSA) guidelines (2019), antibiotic recommendations for the empiric treatment of community-acquired pneumonia (CAP) were based on the following bacterial core respiratory pathogens: *Streptococcus pneumoniae*, *Haemophilus influenzae*, *Mycoplasma pneumoniae, Moraxella catarrhalis*, *Chlamydophila pneumoniae*, *Staphylococcus aureus* and *Legionella species*. Moreover, ATS/IDSA recommend that each acute care center should develop its own validated screening tools for methicillin-resistant *S. aureus* (MRSA) and Pseudomonas based on local risk factors identified by the stewardship team. Not including the immunocompromised, as per the European guidelines (2011), ATD/IDSA (2019) recommends a non-anti-pseudomonal beta-lactam, plus or minus a macrolide or respiratory fluoroquinolone, as empiric therapy for CAP, not covering MRSA and *Pseudomonas aeruginosa* (PA) [1,2,3,4,5]. In both immunocompetent and immunocompromised patients, empirical therapy has to be expanded beyond the core respiratory pathogens, such as MRSA and resistant gram-negative bacilli (including PA), when risk factors for drug-resistant organisms or opportunistic pathogens are present (such as previous hospitalization, exposure to a parenteral antibiotic in the last 90 days or previous cultures of MRSA or Pseudomonas from the airways) [6]. However, in the last few decades, bacteria considered typical of hospital settings, including PA, are also gaining importance in the community [2,7]. CAP caused by PA (PA-CAP) in healthy and non-comorbid patients has been reported in the literature and is associated with high mortality; however, data are limited, and the exact incidence, pathogenesis and clinical features are not entirely known. Rare PA-CAP case reports described the genetic features of the isolated PA. Here, the description of a case of PA-CAP with a rapid and fatal outcome in a healthy young man was reported, together with the genomic characterization of the PA strain isolated from the patient. The case description is followed by a detailed literature review on PA-CAP.

## 2. Materials and Methods

We performed a comprehensive literature review, which included publications from 2010 to 2022, via an online literature search using PubMed and the following keywords: “*Pseudomonas aeruginosa* and community acquired pneumonia”.

Culture and identification methods. Blood cultures were performed using Plus Aerobic/F, Plus Anaerobic/F culture vials incubated in the automated Bactec FX system (Becton Dickinson, East Rutherford, NJ, USA). A Direct Gram stain was performed on each positive blood culture bottle, then subcultured on solid agar media. Respiratory sample was analyzed by a semi-quantitative culture, seeded on different media and incubated at 37 °C for up to 48 h. The identification of the isolates was performed using the VITEK MALDI TOF MS (bioMerieux); antimicrobial susceptibility testing of strains was performed using the Vitek 2 automated system (bioMerieux, Marcy l’Etoile, France). Minimal inhibitory concentrations (MIC) were interpreted using EUCAST clinical breakpoints.

Whole Genome Sequencing. Genomic DNA of the PA strain was extracted with DNeasy Blood and Tissue kit (Qiagen, Hilden, Germany), sequencing libraries were prepared using DNA Prep (M) Tagmentation (Illumina, San Diego, CA, USA) and run with a Miseq platform (Illumina) producing pair-end reads (2 × 250 bp). Reads were evaluated for quality using FastQC, species and possible contamination were checked through Kraken2 [8,9]. Raw reads were filtered with Trimmomatic ver. 0.39 and assembled by using Unicycler ver. 0.4.8 [10,11]. The assembly quality was evaluated by QUAST ver. 5.0.2 [12]. In silico Multi Locus Sequence Type (MLST) was determined using the Pasteur BIGSdb for PA [13]. The presence of plasmids was evaluated by using PlasmiFinder and visualizing the graph of the assembly through Bandage [11,14]. Research of genes encoding virulence factors was performed through VFanalyzer, querying the virulence factor database (VFDB) for PA. A deeper analysis was carried out with ABRICATE, querying the entire VFDB (32,164 genes, downloaded on 6 December 2022) and searching for virulence genes acquired from other bacterial species. The virulence factor profile was compared to those of the four PA reference genomes (PAO1, PA14, LESB58, PA7) [4,15,16,17,18].

## 3. Case Report Description

On 12 March 2022, a 53-year-old man admitted to the Emergency Department of Modena Hospital, Italy, presented right subscapular pain lasting for six hours (not relieved by paracetamol and ketoprofen), followed by dyspnea, fever, cough and hemoptysis. His past medical history was unremarkable, except for the paucisymptomatic SARS-CoV-2 infection experienced roughly two months prior. Vital signs at the arrival revealed a temperature of 36 °C, blood pressure of 120/60 mmHg, heart rate of 120 bpm, oxygen saturation (SpO_2_) of 90% at room air, with a slightly increased respiratory rate (20 breaths per minute). Electrocardiogram documented a sinus rhythm. Blood exams reported only hyponatremia (132 mEq/L, range 136–146 mEq/L), slightly high levels of C-reactive protein (CRP, 2 mg/dL, range 0–0.7 mg/dL), alteration of D-dimer (3608 ng/mL, range 0–500 ng/mL), significant elevation of procalcitonin level (PCT, 51 microg/L, normal range < 0.5 microg/L) and normal lactates value (1.2 mmol/L) (Table 1).

Arterial blood gas analysis documented hypoxia (pO_2_ 52 mmHg, range 75–100 mmHg, SpO_2_ 90%, range 95–99%), so that an oxygen therapy via a Venturi mask FiO_2_ 35% was implemented, with achievement of SpO_2_ 95%. Chest X-ray displayed a right upper lobe opacification (Figure 1).

Considering the thoracic pain and D-dimer elevation, CT angiography was performed: it confirmed the right upper lobe opacification with a suspected air bubble inside, excluding pulmonary embolism or aortic dissection (Figure 1). The patient was admitted to the Internal Medicine Unit: considering the reported penicillin allergy, empiric treatment with intravenous levofloxacin 750 mg/day was started.

Six hours after admission, the patient presented acute neurological deterioration with a stuporous state and mottled skin; arterial blood gas analysis showed respiratory and lactic acidosis (pH 7.1 mmHg, PaO_2_ 45 mmHg, PaCO_2_ 42 mmHg, HCO_2_- 16.5 mmol/L, BE −15.8 mmol/L, SpO_2_ 73.3% in oxygen therapy FiO_2_ 100%, lactates 9.7 mmol/L). He was transferred to the Intensive Care Unit where endotracheal intubation was complicated by cardiac arrest. Cardiopulmonary resuscitation was performed, followed by pneumothorax, so that a chest drain was placed. Twenty-four hours after admission, a repeat chest X-ray revealed complete right upper lobe, right lower and left mid-basal opacification, with diffuse thickening of the bilateral lower-middle lung (Figure 1). Since the clinical picture was compatible with multi-organ failure and septic shock, bronchoalveolar lavage was performed and empiric antimicrobial therapy was improved by adding cefepime 6 g/day and linezolid 600 mg × 2/day. However, the clinical picture evolved in irreversible metabolic acidosis and multi-organ failure, confirmed by blood exams (Table 1). The patient died 36 h after admission. The autopsy findings confirmed a cavitated lesion (2.5 cm) at the right upper lung lobe, surrounded by parenchyma of increased consistency and covered by fibrinous induration, with pleural involvement; left lung parenchyma increased consistency at the lower lobe and was covered by fibrinous induration at the upper lobe. There was also bilateral diffuse alveolar hemorrhage and pulmonary oedema.

Microbiological investigations revealed that blood and bronchoalveolar lavage cultures were positive for PA with a good sensitivity profile (susceptible to meropenem MIC 1 µg/mL, amikacin MIC 4 µg/mL, ceftazidime/avibactam MIC 2 µg/mL and ceftolozane/tazobactam MIC 0.5 µg/mL, intermediate to piperacillin/tazobactam MIC 8 µg/mL, ceftazidime MIC 2 µg/mL, cefepime MIC 2 µg/mL and ciprofloxacin MIC 0.25 µg/mL). The whole genome of the isolated PA strain was sequenced: the PA strain belongs to ST1184, a rare ST, not listed among PA high-risk clones [18]. The comparison of the virulence factor profile of our PA strain with that of reference genomes (PAO1, PA14, LESB58, PA7) showed the same virulence profile of PAO1. PAO1 is a reference genome that, together with PA14, is classified as a “classical” strain, possessing the well-studied virulence determinant Type III Secretion System (T3SS) and its related effector gene (exoS/exoU), which are lacking in PA7-like PA and considered “clonal outlier” strains, with a reduced virulence [19,20,21,22]. The isolated strain carried the exoenzyme gene exoS, a type III translocated effector involved in virulence, while the alternative and the effector typically present on the PAPI-2 pathogenicity island of the PA14 genome, exoU, was absent. The wild type of form of the ladS gene, whose mutation contributes to the increased cytotoxicity of PA14, was confirmed [23,24]. Exolysin genes exlA and exlB were absent in the studied strain. An extensive analysis querying the entire VFDB did not reveal the presence of additional virulence factors acquired from other species besides those available in the PA database. The analysis of the plasmid content showed the absence of any plasmids.

## 4. Discussion and Literature Review

The multinational prevalence of PA-CAP was estimated to be about 4%, specifically 3.8% in Europe (about 1.4% in Italy, where the case described above was from), 4.3% in North America, 5.2% in Asia, 4.9% in South America, 5.5% in Africa and 3.1% in Oceania [7,25].

### 4.1. Previous PA-CAP Cases Description Review

Rare cases of PA-CAP in healthy patients have been described since the 1960s: Hatchette et al. analyzed all cases indexed in PubMed from 1966 to 2000 (*n* = 11) [26], while Maharaj et al. analyzed cases indexed in PubMed from 2001 to 2016 (*n* = 9) [27]. We undertook a review of PA-CAP in immunocompetent and healthy patients described from 2010 to 2022, summarized in Table 2.

On review of the 12 cases reported, there was male gender predisposition with 4 females and 8 males affected, mean age of 35.33 years (1–67 years). A total of 33% (4/12) were active smokers, and only one case presented alcohol abuse. In total, 42% (5/12) had previous environmental exposure (contaminated water and metalfluid). Upon admission, 58% (7/12) reported pleuritic chest pain, together with right back pain in 71% (5/7). Other symptoms were cough (91%, 11/12), fever (75%, 9/12), hemoptysis (50%, 6/12), dyspnea (50%, 6/12), diarrhea (8%, 1/12), weight loss (8%, 1/12) and diffuse bone pain (8%, 1/12). Concerning white blood cells (WBC), 42% (5/12) presented leukocytosis, while 42% (5/12) had leucopenia. In relation to radiological features, 83% (10/12) showed right pneumonia, with upper lobe consolidation in 58% (7/12) and evident cavitation in 33% (4/12). The left lung was interested only in 25% (3/12). In addition to the respiratory samples, PA was also isolated from blood in 58% (7/12) and pleural effusion in 16% (2/12). In two cases, co-infection with Influenza A occurred, while in other two cases, other bacteria were isolated from respiratory samples: *Acinetobacter baumanii* and *Klebsiella pneumoniae.* Most cases started with empirical antibiotic therapy that did not perfectly cover PA and were subsequently modified accordingly with the identification of the pathogen. The duration of therapy varied from 14 to 30 days. Of the reported cases, 75% (9/12) needed intensive care and 33% (4/12) died: three cases died after a few hours from the admission (median time 11.3 h), and another one died after 9 days. Among survivors, 66% (8/12) were discharged after a median recovery of 23,75 days (7–35 days); the median recovery of patients discharged after intensive care (42%, 5/12) was 30 days.

### 4.2. Risk Factors and Score for PA-CAP Identification

Lung structural abnormalities, immunocompromised condition, chronic heart failure, cerebrovascular disease, burns, malnutrition, prolonged antibacterial therapy, recent hospitalization (in the past 90 days), previous PA infection/colonization and premature birth were reported as risk factors for PA-CAP. In people without these underlying diseases, advanced age, smoking, alcohol use and exposure to contaminated liquids were considered [7,28,30,34,40,41,42,43,44,45,46]. A few reported cases were attributed to an identifiable source: contaminated aerosolized water from a home humidifier, hot tub water, water in a rearing beetles cage, washrooms in care facilities and factory metalworking fluid [29,30,34,35]. Welders and foundry workers have to be considered workers at risk of PA-CAP since aqueous oil emulsions used in metal processing are frequently contaminated or colonized by Gram-negative bacteria [47,48,49,50]. Recently, Influenza A has been considered a risk factor for PA infection: influenza viruses contribute to the dysfunction and death of respiratory epithelial cells, allowing bacterial adherence and invasion [36,38,51]. In the case described above, the patient presented paucisymptomatic SARS-CoV-2 infection two months before. We suppose SARS-CoV-2 infection could induce temporary immunosuppression or dysfunction of respiratory epithelial cells, increasing susceptibility to subsequent PA superinfection. Many cases had no specific identifiable risk factors, as in the case described above: the use of scores could help for the correct diagnosis in these situations [26]. CURB-65 and Pneumonia Severity Index (PSI) are the most known and used scores for pneumonia, and they are usually higher in PA-CAP than for pneumonia caused by other pathogens. However, in 2015, Prina et al. [52] proposed a new PES score (*Pseudomonas aeruginosa*, extended-spectrum β-lactamase Enterobacterales and methicillin-resistant Staphylococcus aureus) more promising for detecting PA-CAP, even in the immunocompetent host, independent of the site of pneumonia acquisition [40,52,53]. It is easily calculable at the bedside, and also, if it cannot discriminate between PA and extended-spectrum b-lactamase Enterobacterales or MRSA, incorporated into a clinical algorithm, it could become a valuable tool for improving therapeutic choice. According to this score, later validated in Japan, patients with ≥2 risk factors for PES pathogens should be treated with an appropriate therapy for hospital-acquired pneumonia; indeed, our case PES score was 2 [3,52,53,54,55]. Nevertheless, to date, no validated scoring systems exist with a sufficiently high positive value to identify patients at risk of PA-CAP only to target anti-pseudomonal beta-lactam treatment. Moreover, although these different scores may be used as a first step in a wider strategy, clinical features and subsequent advanced diagnostic tests are necessary for a correct diagnosis.

### 4.3. Clinical Features of PA-CAP

Patients generally have no specific symptoms such as cough, fever and dyspnea; hemoptysis and pleuritic chest pain, particularly in the right back, should induce to consider PA infection, as reported in the cases above [28,29,30,32,33,34,35]. Empyema and abscess in other sites, such as in the brain, could accompany lung parenchyma involvement [29,37,38]. Concerning laboratory features, variable leukocyte count is reported: patients with normal values could have relatively contained infection in the lungs. PCT could be negative since its transcription is probably not stimulated by localized PA necrotizing pneumonia, even if clinically and radiologically advanced. In contrast, high CRP levels have been reported to have the best discriminatory performance for the severity of the infection [27]. In the case described above, upon admission, the leukocyte count was normal, but high PCT and CRP levels suggested a systemic infection; in patients with CAP, the rate of true positive blood cultures is <10% [56]. In relation to imaging features, bilateral infiltrates, cavitary pneumonia, nodular infiltrates and pleural effusion are more commonly described [31,57]. Previous literature reported that PA-CAP involved the right upper lobe (RUL) in two-thirds of patients [26]. Considering the case reported above and the reviewing cases, 61% presented RUL involvement. Radiological features are usually characterized by right upper lobar opacities in chest X-rays and lobar pneumonia with a cavity in chest CT-scans. However, in the early time of the disease, necrotizing pneumonia can appear only as consolidation in the chest radiography, leading to an underestimation of the degree of parenchymal destruction [27]. Two different presentations of PA-CAP are described in autopsy: diffuse subpleural hemorrhagic nodular lesions with occasional necrotic parenchyma; and necrotic lung lesions within the lobular boundaries, including multiple necrotic nodules and hemorrhagic parenchyma [58]. In reviewed cases, only two autopsies were reported, both described as the second presentation, together with the case described above [33,34]. PA-CAP is associated with possible Multi Organ Failure (MOF) and a 30-day mortality rate of 33–63%, generally occurring with a median time of 11 h from admission to death, according to the reviewed cases [26,27,40]. It is still unclear how such severe CAP developed into septic shock and MOF in healthy young men, such as in the case described above [30]. At the same time, despite inadequate empiric antimicrobial treatment and intensive cure necessity, some patients were discharged without follow-up after a median recovery of 30 days [29,30,37,38,39]. The different outcome suggests the need to better investigate the microbiological characteristics of this pathogen in relation to the different host responses.

### 4.4. Microbiology and Whole Genome Sequencing

Genomic characterization of PA-CAP strains isolated from patients was only reported in a few cases. Huhulescu et al. (2011) [28] reported a fatal PA-CAP case due to a PA strain belonging to ST313, but whole genome sequencing was not performed. Woods et al. (2017) [35] described a PA-CAP case linked to the use of a humidifier, comparing the PA isolates obtained from the humidifier basin and from the bronchoscopy tissue of the patient through Whole Genome Sequencing. The identification of the ST and virulence profile of PA isolates reported by Woods et al. that we performed to compare these PA isolates with our isolate, revealed that they belonged to a different ST (ST1182) and showed a different virulence profile (similar to that of PA14). The genomic investigation of a citywide PA outbreak in South Africa, conducted by Opperman C. J. et al. (2022) [59] reported that CAP cases were due to PAs from different STs (ST1051, ST412, ST709 and ST303) and with different virulence profiles. All of them belonged to “classical” strain (“PAO1” and “PA14”) groups, possessing the T3SS and related effector genes (exoS or exoU). Altogether, these data already showed that PA isolates with different STs and different virulence profiles were responsible for CAP. Additionally, our analysis, performed on the genome of the strain here reported, revealed that the isolate possesses the same virulence profile of PAO1, a reference genome that, together with PA14, is classified as a “classical” strain possessing the well-studied virulence determinant Type III Secretion System (T3SS) and its related effector gene (exoS), that lacks in PA7-like PA, considered “clonal outlier” strains [20,21,22]. Exolysin genes exlA and exlB, generally associated with PA7, were absent in the studied strain, as in PAO1 and PA14 genomes [60]. The bioinformatic analysis did not reveal the presence of other additional virulence genes, possibly inherited from other species as often observed in ESKAPE pathogens similar to PA. The presence of other factors, such as a mutation in virulence genes or the presence of new uncharacterized virulence genes, which allow an increased virulence, cannot be excluded. Our study reported an additional PA isolate, which is responsible for CAP, with a different ST and a different virulence profile with respect to already known PA isolates associated with CAP.

### 4.5. Antibiotic Therapy

In recent years, empirical treatment for patients with CAP has become challenging due to the emerging prevalence of drug-resistant bacteria in community settings [7,17]. Many hospitalized patients receive empirical antipseudomonal coverage while waiting for 48–72 h for specific pathogen identification and antibiotic susceptibilities [7,61]; if PA was not isolated, discontinuation of this coverage was recommended. However, this practice promoted antibiotic overuse with the risk of inducing antimicrobial resistance. At the same time, the delay between initial diagnosis and availability of antibiotic susceptibilities could negatively affect outcomes in patients with PA-CAP due to inappropriate antibiotic coverage [7]: for patients with suspected PA-CAP, combination antibiotic therapy should be administered within an hour [37]. Clinical practice guidelines emerged to improve the quality of care and standardization of patients’ management with CAP [62]. In the 2016 ATS/IDSA guidelines for hospital-acquired pneumonia (HAP) and ventilator-associated pneumonia (VAP), the term healthcare-associated pneumonia (HCAP) was eliminated in favor of the dichotomy of CAP or HAP, leading to less broad-spectrum antimicrobial use and anti-PA agents for empiric treatment for CAP previously considered HCAP.

Both the European guidelines (EG) and ATD/IDSA guidelines recommend empiric therapy with a non-anti-pseudomonal beta-lactam and a macrolide or a quinolone alone in non-severe inpatient pneumonia while recommending a non-anti-pseudomonal beta-lactam, plus either a macrolide or a quinolone, in severe inpatient pneumonia for patients without risk factors for PA or MRSA [1,2,3]—the same recommendation provided by The National Institute for health and care excellence (NICE) [63]. Given the current low prevalence of PA-CAP, only a small subgroup of patients may really require empiric antipseudomonal antibiotic coverage [7]: ATS/IDSA recommends initiating antipseudomonal drugs as an empiric treatment after considering the local epidemiology and validated risk factors for PA-CAP, such as prior respiratory isolation of Pseudomonas, recent hospitalization or the use of parenteral antibiotics in the last 90 days. In these cases, an antipseudomonal beta-lactam plus a fluoroquinolone or aminoglycoside is recommended [1,2]. In addition, specific guidelines about CAP in immunocompromised patients have been recently published: the initial empirical therapy for immunocompromised patients should cover resistant gram-negative bacilli, including PA, only if there is a history of colonization or infection with a resistant gram-negative bacilli in the prior 12 months, previous hospitalization with exposure to broad-spectrum antibiotics, the presence of a tracheostomy, neutropenia, chronic corticosteroid therapy or a history of pulmonary comorbidity [6]. It is important to point out that less than one-third of the recommendations in the CAP guidelines were based on evidence derived from high-quality studies, and most recommendations remain based on low-level evidence [62]. Controlled studies are needed to study the role of inadequate empiric antibiotic therapy, especially in severe CAP.

### 4.6. Should the Guidelines Be Changed?

The CAP official clinical practice guideline approved by ATS/IDSA and the subsequently published Consensus Statement for treatment of immunocompromised patients with suspected CAP suggested extending empirical therapy beyond core respiratory pathogens when (1) risk factors for drug-resistant organisms and (2) a prior identification of MRSA or PA in the respiratory tract. According to these recommendations, initial empirical therapy should cover PA if there is a history of colonization or infection with a resistant gram-negative bacillus in the prior 12 months, previous hospitalization with exposure to broad-spectrum antibiotics, the presence of a tracheostomy, neutropenia, or a history of pulmonary comorbidity (i.e., cystic fibrosis, bronchiectasis, or recurrent exacerbations of COPD requiring glucocorticoid and antibiotic use) [2].

Our case seems to suggest the need for a revision of the post-pandemic SARS-CoV2 epidemiological criteria by including the ‘previous COVID-19 disease’ as a possible risk factor for PA-CAP. Moreover, our fatal outcome underlines the necessity of further study to investigate the effectiveness and safety of using illness severity tools as decision support to guide the intensity of treatment in adults hospitalized for pneumonia.

Indeed, the delay in empirical antimicrobial therapy will place the patient at an increased risk of mortality, and patients transferred to an ICU after admission to a hospital ward experience higher mortality than those directly admitted to the ICU from an emergency department [61,62,63]. This higher mortality may in part be attributable to progressive pneumonia, but “mis-triage” of patients with unrecognized severe pneumonia may be a contributing factor, such happened in our case [61]. The current use of new scores, such as PES over PSI and CURB-65 (the only ones mentioned in the available guidelines), together with physician judgment could guide not only the site-of-care but also the need for empiric extended-spectrum antibiotic treatment. We hope that future research on new illness severity scores will help clinicians guide the intensity of treatment in adults hospitalized for pneumonia focusing on pathogens.

## 5. Conclusions

Although PA-CAP is rare in immunocompetent patients, clinicians should consider this pathogen in their differential etiological diagnosis for CAP since, after an insidious course, it is characterized by necrotizing pneumonia with high morbidity and mortality. Risk factors identified for PA-CAP in healthy young people, such as smoking, alcohol use and contaminated water exposure, together with atypical clinical signs (hemoptysis and right back pain) and radiological features frequently described (right upper lobe opacification), should guide clinicians to consider PA in patients with CAP. WBC and PCT use are not reliable in the management of localized necrotizing or cavitary pneumonia. The PES score may help to address the use of appropriate empiric therapy in high-risk patients and consider rigorous antibiotic de-escalation whenever possible as part of a responsible antimicrobial stewardship policy. Analysis of this CAP case, together with the comparison of the other PA-CAP case described in the literature, shows that PA isolates with different STs and different virulence profiles, could be responsible for CAP with a rapidly fatal course. Finally, our case underlined that co-infection with a respiratory virus, in particular, with SARS-CoV-2 infection, deserves future attention. Considering the high severity and rapid progression of PA-CAP, further studies are needed to identify sources of infection, unknown risk factors, microbiological features and immunological determinants. These factors could lead to a more selective and earlier identification of patients at risk for PA-CAP in order to better define a target treatment algorithm.

## Figures and Tables

**Figure 1 microorganisms-11-01112-f001:**
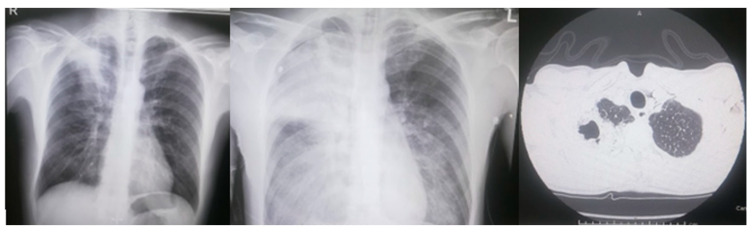
On the left: chest X-ray upon admission, with right upper lobe opacification. In the middle: chest X-ray 24 h after admission, with complete right upper lobe, right lower and left mid-basal opacification, diffuse thickening of bilateral lower-middle lung. On the right: right upper lobe opacification with a suspected air bubble inside at the CT angiography.

**Table 1 microorganisms-11-01112-t001:** Blood exams alterations observed upon admission (T1) and 35 h after admission (T2). LEU = leucocytes. N = neutrophils. PLT = platelets. CRP = C-reactive protein. PCT = procalcitonin. ALT = alanine transaminase. CREA = creatinine. eGFR = estimated glomerular filtration rate LDH = lactate dehydrogenase. XDP = D-dimer. CK = creatine kinase. TRO = troponin. LIP = lipase.

	LEU /mmc	N %	PLT /mmc	CRP mg/dL	PCT µg/L	ALT U/L	CREA mg/dL	eGFR mL/min	LDH U/L	XDP ng/mL	CK U/L	TRO ng/L	LIP U/L
**T0**	4250	91	175,000	2.0	51	19	0.95	91	328	3608	-	-	-
**T1**	1020	35	56,000	7.9	147	1609	4.54	13.7	12 052	2646	80,000	1070	425

**Table 2 microorganisms-11-01112-t002:** PA-CAP cases reported in healthy people from 2010 to 2022. WBC =white blood cell. CRP = C-reactive protein. PCT = procalcitonin.

Reference Site (Year)	Exposure	Sex Age	Smoke Alcohol	Symptoms on admission	Radiology	WBC (/µL) CRP (mg/dL) PCT (ng/mL)	Positive colture for PA Other Infection	Antibiotic Therapy	Course and Outcome
Huhulescu et al [28] Austria (2010)	Hot tub water	F 49	Yes -	Cough Pleuritic chest pain	Left lung infiltration	WBC 3 940 CRP 38.1	Respiratory sample Blood	- Piperacillin/sulbactam - Moxifloxacin	- Intensive care - death after 9 days from admission
Okamoto et al [29] Japan (2011)	Water in rearing beetles cage	F 39	Yes Yes	Cough Pleuritic chest pain Hemoptysis Dyspnea Diarrhea	Right upper consolidation Pleural effusion	WBC 21 500 CRP 6.61	Respiratory sample Blood Pleural fluid	- Ceftriaxone (10 h) - Meropenem, ciprofloxacin (29 days)	- Intensive care - Recovery of 30 days
Kunimasa et al. [30] Japan (2012)	Washrooms, bathroom in care facility	M 25	Yes -	Cough Fever Right back pain Hemoptysis	Right upper consolidation, cavitation	WBC 7 800 CRP 14	Respiratory sample	- Ampicillin, levofloxacin (2 days) - Meropenem, levofloxacin (26 days)	- Intensive care - Recovery of 28 days
Gharabaghi et al. [31] Iran (2012)	Not known exposure	M 26	No -	Cough Fever Bone pain	Left upper consolidation, cavitation, left lower lobe mass with necrosis	WBC normal	Respiratory sample Lung biopsy *Klebsiella pneumoniae*	- Ofloxacin (7 days) - Ciprofloxacin (2 weeks)	-
Fujii et al. [32] Japan (2012)	Not known exposure	M 29	No	Fever Right back pain	Right upper consolidation, cavitation	WBC 26 400 CRP 20	Respiratory sample Blood	- Piperacillin (19 days) - Levofloxacin (1 week) - Piperacillin/tazobactam - Tobramicin (4 weeks) - Ciprofloxacin (2 weeks)	- 2 relapses (after 2 days, after 1 month) - Recovery of 19 days (first time), not recovering (second time), recovery of 28 days (third time)
Takajo et al. [33] Japan (2013)	Not known exposure	F 50	- -	Cough Fever Right back pain Dyspnea Hemoptysis	Right upper lobe consolidation Pneumothorax	WBC 2 100	Respiratory sample Blood	- Meropenem	- Intensive care - Death after 5 h from admission
Campos et al. [34] Brazil (2014)	Metalworking fluid	M 44	Yes -	Cough Fever Right back pain Hemoptysis	Right upper and lower consolidation Left consolidation	WBC 2 880	Respiratory sample Blood	- Ceftriaxone, chlaritromycin	- Intensive care - Death after 7 h from admission
Woods et al. [35] USA (2017)	Home humidifier water	M 30	No No	Cough Fever Right back pain Weight loss	Right upper consolidation, cavitation	WBC 11 300	Respiratory sample	- Ceftazidime (7 days) - Ciprofloxacin (14 days)	- Recovery of 7 days
Su et al. [36] Taiwan (2018)	Not known exposure	F 39	-	Cough Fever Dyspnea	Bilateral patchy infiltrates	WBC 810	Respiratory sample Blood *Influenza A(H1N1)pdm09*	- Piperacillin/Tazobactam	- Intensive care - Death after 23 h from admission
Wang et al. [37] China (2019)	Not known exposure	M 25	No No	Cough Dyspnea Hemoptysis Brain abscess	Multiple right upper and lower consolidations Pleural effusion	WBC 390 PCT 100	Respiratory sample Blood	- Levofloxacin (1 day) - Linezolid, meropenem (1 day) - Levofloxacin, meropenem (5 days) - Ceftazidime, ciprofloxacin (14 days)	- Intensive care - Recovery of 28 days
Dong et al. [38] China (2019)	Not known exposure	M 1	No No	Cough Fever Dyspnea Hemoptysis Bloody pleural effusion	Right middle and lower lobe consolidation, cavitation Pleural effusion	WBC 41 780 CRP 102 PCT 4.49	Respiratory sample Pleural fluid *Influenza A*	- Cefaclor (1 day) - Ceftriaxone (1 day) - Meropenem (8 days) - Cefoperazone/sulbactam, linezolid (2 days) - Cefoperazone/sulbactam, levofloxacin (20 days)	- Intensive care - Recovery of 32 days
Wang et al. [39] China (2019)	Not known exposure	M 67	Yes (past) -	Cough Fever Dyspnea	Right upper consolidation, cavitation	WBC 20 360	Respiratory sample *Acinetobacter baumanii*	- Imipenem/cilastatin, linezolid - moxifloxacin (3 days) - Polimixin b, amikacin (30 days)	- Intensive care - Recovery of 35 days

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
