# Peer review of "A Fatal Case of Pseudomonas aeruginosa Community-Acquired Pneumonia in an Immunocompetent Patient: Clinical and Molecular Characterization and Literature Review"

_microorganisms, 2023, doi:10.3390/microorganisms11051112_

Round 1

Reviewer 1 Report

This paper did a literature review of clinical and molecular characteristics of PA-CAP in the past 13 years, overall the paper is well written. It is suggested that the authors give more details about the possible CAP guidelines changes, which will provide more insights concerning this topic.

Author Response

Dear reviewer 1,

We are very grateful for your constructive comments and suggestions to our paper entitled: A fatal case of Pseudomonas aeruginosa community-acquired pneumonia in an immunocompetent patient: clinical and molecular characterization and literature review”

We here provide a point-by-point reply to the comments, and we have incorporated the related changes in the manuscript.

We thank you for your thoughtful insights which helped to significantly improve the manuscript.

- “This paper did a literature review of clinical and molecular characteristics of PA-CAP in the past 13 years, overall, the paper is well written.

It is suggested that the authors give more details about the possible CAP guidelines changes, which will provide more insights concerning this topic.”

Thanks, for your important suggestion, according to your suggestion we added a paragraph discussing possible CAP guidelines changes in light with our literature revision and our clinical case (line 330-356)

“4.5 Should the guidelines be changed?

The CAP official clinical practice guideline approved by ATS/IDSA and the subsequently published Consensus Statement for treatment of immunocompromised patients with suspected CAP, suggested extending empirical therapy beyond core respiratory pathogens when (1) risk factors for drug-resistant organisms and (2) a prior identification of MRSA or PA in the respiratory tract. According to these recommendations, initial empirical therapy should cover PA if there is a history of colonization or infection with a resistant gram-negative bacillus in the prior 12 months, previous hospitalization with exposure to broad-spectrum antibiotics, the presence of a tracheostomy, neutropenia, or a history of pulmonary comorbidity (i.e. cystic fibrosis, bronchiectasis, or recurrent exacerbations of COPD requiring glucocorticoid and antibiotic use).

Our case seems to suggest the need for a revision of the post-pandemic SARS-CoV2 epidemiological criteria by including the 'previous Covid 19 disease' as a possible risk factor for PA-CAP. Moreover, our fatal outcome underlines the urge of further study investigate the effectiveness and safety of using illness severity tools as decision support to guide the intensity of treatment in adults hospitalized for pneumonia.

Indeed, the delay in empirical antimicrobial therapy will place the patient at increased risk of mortality and patients transferred to an ICU after admission to a hospital ward experience higher mortality than those directly admitted to the ICU from an emergency department [64-67]. This higher mortality may in part be attributable to progressive pneumonia, but “mis-triage” of patients with unrecognized severe pneumonia may be a contributing factor, such happened in our case [64]. The current use of new scores such as PSE over PSI and CURB-65 (the only ones mentioned in the available guidelines) together with physician judgment could guide not only the site-of-care but also the need for empiric extended-spectrum antibiotic treatment. We hope that future research on new illness severity score will help clinicians guide the intensity of treatment in adults hospitalized for pneumonia focusing on pathogen.”

Moreover, the reviewer's suggestion encouraged us to add a brief part about the PES score (line 212-228):

“CURB-65 and Pneumonia Severity Index (PSI) are the most known and used scores for pneumonia and they are usually higher in PA-CAP than for pneumonia caused by other pathogens. However, in 2015 Prina et al. [55] proposed a new PES score (Pseudomonas aeruginosa, extended-spectrum β-lactamase Enterobacterales, and methicillin-resistant Staphylococcus aureus) more promising for detecting PA-CAP even in the immunocompetent host, independent of the site of pneumonia acquisition [42, 54, 55]. It is easily calculable at the bedside and also if it cannot discriminate between PA and extended-spectrum b-lactamase Enterobacterales or MRSA, incorporated into a clinical algorithm could become a valuable tool for improving therapeutic choice. According to this score, later validated in Japan, patients with ≥2 risk factors for PES pathogens should be treated with an appropriate therapy for hospital-acquired pneumonia, indeed our case PES score was 2 (3, 55-57]. Nevertheless, to date no validated scoring systems exist with sufficiently high positive predictive value to identify patients at risk of P. aeruginosa-CAP only to target anti-pseudomanal betalactam treatment. Moreover, although these different scores may be used as a first step of a wider strategy, clinical features and subsequent advanced diagnostic tests are necessary for a correct diagnosis.”

Finally we consider and add the following reference (line 538):

“57. Cillóniz C, Dominedò C, Nicolini A, Torres A. PES Pathogens in Severe Community-Acquired Pneumonia. Microorganisms. 2019 Feb 12;7(2):49.”

Reviewer 2 Report

Barp et al. present a case report of a young immunocompetent male patient that had a fatal case of CAP caused by Pseudomonas aeruginosa. Following the case report, the authors outline other publications that describe cases of CAP that were caused by P. aeruginosa to identify common risk factors and clinical presentations among the patients. The involvement of P. aeruginosa with CAP is an important topic clinically that I believe will be of interest to the readers of the journal. The manuscript is relatively well-written, and I believe the authors’ evaluation of PA and CAP will add meaningfully to the literature; however, I believe the manuscript may be improved prior to publication by improving the clarity of certain discussion topics. I also believe the manuscript will benefit from some proofreading to clean up a few of the grammatical inconsistencies. My specific comments are as follows:

·         Line 49 – 50 – The guidelines also specify that culturing MRSA or Pseudomonas from the airways is a major risk factor for such difficult-to-treat pathogens.

·         Introduction – I think it is worth mentioning that the ATS/IDSA recommend that each acute care center develop their own validated screening tools for MRSA and Pseudomonas based on local risk factors identified by the stewardship team.

·         Lines 87 – 88 – Recommend modifying to include the country: Modena Hospital, Italy, for right…

·         Line 90 – do we know which variant of COVID-19 the patient had?

·         Table 1 – was the patient’s lactic acid within the normal limit upon admission? Lactate is reported 6 hours later as 9 mmol/L.

·         Seeing as levofloxacin is renally eliminated, can we please have the patient’s calculated creatinine clearance, GFR, or serum creatinine reported at admission and 35 hours after admission as well?

·         Line 128 – out of curiosity, is there a reason why the medical team chose linezolid as the anti-MRSA agent instead of vancomycin?

·         Lines 136 – 152 – In addition to the pathogen’s virulence profile, can the authors please describe the susceptibility profile or the isolate? For example, was the Pseudomonas susceptible to levofloxacin, the antibacterial that was initially selected for the patient?

·         Line 155 – When the authors parenthetically state (1-1, 4% in Italy…), it is not clear to me what the “1-1” values signify.

·         Lines 187 – 192 - The authors list several proposed risk factors for Pseudomonas involvement with CAP, but Table 2 seems to only address a few risk factors such as smoking and alcohol use. Did the authors review the cases in their literature search to see how many of the patients possessed risk factors such as being immunocompromised or recently hospitalized?

·         Line 209 – the order Enterobacterale is typically used instead of the family Enterobacteriaceae after the somewhat recent reclassification of the organisms within the order.

·         Line 217 – are the authors trying to say that procalcitonin may or may not be elevated? The phrase “Quite positive” sounds like the procalcitonin will be highly elevated, but the rest of the sentence seems to describe why the procalcitonin may not be very high. I recommend rephrasing that sentence for clarity.

·         Lines 283 – 287 – I think the authors are mischaracterizing the ATS/IDSA recommendations for CAP slightly. For non-severe inpatient pneumonia, the ATS/IDSA recommends a non-anti-Pseudomonal beta-lactam AND a macrolide, or as an alternative, a respiratory quinolone may be used alone. For severe pneumonia the ATS/IDSA recommends a beta-lactam + either a macrolide or a quinolone. The authors’ description of the guidelines makes it sound like a beta-lactam alone or a beta-lactam plus a respiratory quinolone are recommended options for non-severe inpatient CAP, which is not the case.

·         Instead of referring to the ATS/IDSA guidelines as “American guidelines” I think it is more appropriate to refer to the guidelines by the organizations ATS/IDSA that wrote the guidelines.

·         Line 292 – I think it is more accurate to state that if there is prior respiratory isolation of Pseudomonas, then the ATS/IDSA recommend pseudomonal coverage. The phrase “prior infection” used by the authors is a little vague.

·         Line 293 – 294 – I don’t see where the ATS/IDSA guidelines recommend using double pseudomonal coverage for patients with PA risk factors. In Table 4 of the guidelines, the ATS/IDSA CAP guidelines reference the 2016 HAP/VAP guidelines and recommend anti-PA beta-lactams alone. I don’t believe the 2019 ATS/IDSA CAP guidelines ever mention aminoglycosides in any capacity.

·         It is probably worth mentioning that the 2016 IDSA HAP/VAP guidelines eliminated the term healthcare-associated pneumonia in favor of the dichotomy of CAP or HAP, which has led to less broad spectrum antimicrobial use and presumably less use of anti-PA agents for empiric CAP treatment (which may have previously been considered HCAP instead of CAP).

As I mentioned earlier, I recommend that someone proofread the entire manuscript for grammatical mistakes. An example of some edits I recommend are:

·         Table 2 – the patient in the Huhulescu et al. study in Austria was presumably exposed to hot tub [not tube] water

·         Table 2 - It looks like the E in ceftriaxone was cutoff

·         Lines 179 – 181 – I think the sentence need to be modified grammatically to something like “Most cases started with empiric antibiotic therapy that did not perfectly cover Pa and were subsequently modified..”

·         Lines 183 – 185 – Among survivors, 66% (8/12) [were] discharged after a [median] recovery of 23.75 days (7 – 35 days); the [median] recovery of patients…

·         Line 309 – why is community acquired pneumonia written out if the acronym CAP is used throughout the rest of the manuscript?

·         Line 316 – I do not believe a comma is needed after “PES score”

·         Line 320 – “…could be responsible [for] CAP with a…”

·         Line 322 – “…coinfection with [a respiratory] virus”

Author Response

Dear Reviewer 2,

We are very grateful for your constructive comments and suggestions to our paper entitled: A fatal case of Pseudomonas aeruginosa community-acquired pneumonia in an immunocompetent patient: clinical and molecular characterization and literature review”

We here provide a point-by-point reply to the comments, and we have incorporated the related changes in the manuscript.

We thank you for your thoughtful insights which helped to significantly improve the manuscript.

“Barp et al. present a case report of a young immunocompetent male patient that had a fatal case of CAP caused by Pseudomonas aeruginosa. Following the case report, the authors outline other publications that describe cases of CAP that were caused by P. aeruginosa to identify common risk factors and clinical presentations among the patients. The involvement of P. aeruginosa with CAP is an important topic clinically that I believe will be of interest to the readers of the journal. The manuscript is relatively well-written, and I believe the authors’ evaluation of PA and CAP will add meaningfully to the literature; however, I believe the manuscript may be improved prior to publication by improving the clarity of certain discussion topics. I also believe the manuscript will benefit from some proofreading to clean up a few of the grammatical inconsistencies. My specific comments are as follows:

  • Line 49 – 50 – The guidelines also specify that culturing MRSA or Pseudomonas from the airways is a major risk factor for such difficult-to-treat pathogens.

Thanks, for your suggestion. We will add this information (line 46-52):

In both immunocompetent and immunocompromised patients, empirical therapy has to be expanded beyond the core respiratory pathogens, considering MRSA and resistant gram-negative bacilli (including PA), only if risk factors for drug-resistant organisms or opportunistic pathogens are present (such as previously hospitalization, exposure to parenteral antibiotic in the last 90 days or previous cultures of MRSA or Pseudomonas from the airways)”

  • Introduction – I think it is worth mentioning that the ATS/IDSA recommend that each acute care center develop their own validated screening tools for MRSA and Pseudomonas based on local risk factors identified by the stewardship team.

Thanks, for your suggestion. We will add the sentence in the introduction session (line 40-42)

Moreover ATS/IDSA recommend that each acute care center should develop their own validated screening tools for methicillin-resistant S. aureus (MRSA) and Pseudomonas based on local risk factors identified by the stewardship team.”

  • Lines 87 – 88 – Recommend modifying to include the country: Modena Hospital, Italy, for right…

Thanks, for your suggestion,

 “…Modena Hospital for right…” is changed to “…Modena University Hospital, Italy, for right…”

  • Line 90 – do we know which variant of COVID-19 the patient had?

No, unfortunately, we do not know the variant but based on epidemiological regional surveillance report, Omicron B.1.529 was the main circulating variant in Emilia-Romagna in January 2022.

(https://www.epicentro.iss.it/coronavirus/pdf/sars-cov-2-monitoraggio-varianti-indagini-rapide-17-gennaio-2022.pdf)

  • Table 1 – was the patient’s lactic acid within the normal limit upon admission? Lactate is reported 6 hours later as 9 mmol/L.

On admission, lactic acid level was normal (1.2mmol/l), we add this value as suggested (line 102)

  • Seeing as levofloxacin is renally eliminated, can we please have the patient’s calculated creatinine clearance, GFR, or serum creatinine reported at admission and 35 hours after admission as well?

According to our suggestion we will add the value on table 1.

  • Line 128 – out of curiosity, is there a reason why the medical team chose linezolid as the anti-MRSA agent instead of vancomycin?

A recent systematic review and comparative meta-analysis compared vancomycin and linezolid efficacy against proven MRSA pneumonia; the outcomes were mortality, clinical cure, microbiological evaluation, and adverse events. Seven randomized controlled trials (1239 patients) and eight retrospective cohort or case-control studies (6125 patients) were considered. Clinical cure and microbiological eradication rates were significantly increased in patients treated with linezolid in the randomized controlled trials (clinical cure: risk ratio (RR) = 0.81, 95% confidential interval (CI) = 0.71-0.92; microbiological eradication: RR = 0.71, 95% CI = 0.62-0.81) and in case-control studies (clinical cure: odds ratio (OR) = 0.35, 95% CI = 0.18-0.69). Mortality was comparable between patients treated with vancomycin and linezolid in randomized climical trial (RR = 1.08, 95% CI = 0.88-1.32) and case.control studies (OR = 1.20, 95% CI = 0.94-1.53). Finally, no significant difference in adverse events between vancomycin and linezolid was observed (thrombocytopenia: OR = 0.95, 95% CI = 0.50-1.82; nephrotoxicity: OR = 1.72, 95% CI = 0.85-3.45). According to this meta-analysis we chose linezolid as anti-MRSA agent instead of vancomycin.in severe CAP and in those requiring ICU admission.

Reference: Kato H, Hagihara M, Asai N, Shibata Y, Koizumi Y, Yamagishi Y, Mikamo H. Meta-analysis of vancomycin versus linezolid in pneumonia with proven methicillin-resistant Staphylococcus aureus. J Glob Antimicrob Resist. 2021 Mar;24:98-105. doi: 10.1016/j.jgar.2020.12.009. Epub 2021 Jan 2. PMID: 33401013.

  • Lines 136 – 152 – In addition to the pathogen’s virulence profile, can the authors please describe the susceptibility profile or the isolate? For example, was the Pseudomonas susceptible to levofloxacin, the antibacterial that was initially selected for the patient?

Thanks, for your suggestion. We will add the sentence in the “Case report description” session (line 141-145)

Microbiological investigations revealed that blood and bronchoalveolar lavage culture were positive for PA with a good sensitivity profile (susceptible to meropenem MIC 1ưg/ml, amikacin MIC 4ưg/ml, ceftazidime/avibactam MIC 2ưg/ml and ceftolozane/tazobactam MIC 0.5ưg/ml, intermediate to piperacillin/tazobactam MIC 8ưg/ml, ceftazidime MIC 2ưg/ml, cefepime MIC 2ưg/ml and ciprofloxacin MIC 0.25ưg/ml).”

  • Line 155 – When the authors parenthetically state (1-1, 4% in Italy…), it is not clear to me what the “1-1” values signify.

Sorry it was a typo; “About 1-1,4% in Italy” is changed to “About 1,4% in Italy”

  • Lines 187 – 192 - The authors list several proposed risk factors for Pseudomonas involvement with CAP, but Table 2 seems to only address a few risk factors such as smoking and alcohol use. Did the authors review the cases in their literature search to see how many of the patients possessed risk factors such as being immunocompromised or recently hospitalized?

We appreciate our suggestion however, since the peculiarity of our case is that PA-CAP occurred in an immunocompetent host without known risk factors, we decided to include in our revision only cases of PA-CAP in healthy people, excluding immunocompromised or recently hospitalized.

  • Line 209 – the order Enterobacterale is typically used instead of the family Enterobacteriaceae after the somewhat recent reclassification of the organisms within the order.

According to our right proposal we modified the order with the new term“…and extended-spectrum b-lactamase Enterobacteriaceae or MRSA.” is changed to “…and extended-spectrum b-lactamase Enterobacterales or MRSA.”

  • Line 217 – are the authors trying to say that procalcitonin may or may not be elevated? The phrase “Quite positive” sounds like the procalcitonin will be highly elevated, but the rest of the sentence seems to describe why the procalcitonin may not be very high. I recommend rephrasing that sentence for clarity.

We are completely agree with your proposal “PCT could be quite positive, since its transcription is probably not stimulated by localized PA necrotizing pneumonia, even if clinically and radiologically advanced” is changed to “PCT could be negative, since its transcription is probably not stimulated by localized PA necrotizing pneumonia, even if clinically and radiologically advanced”

  • Lines 283 – 287 – I think the authors are mischaracterizing the ATS/IDSA recommendations for CAP slightly. For non-severe inpatient pneumonia, the ATS/IDSA recommends a non-anti-Pseudomonal beta-lactam AND a macrolide, or as an alternative, a respiratory quinolone may be used alone. For severe pneumonia the ATS/IDSA recommends a beta-lactam + either a macrolide or a quinolone. The authors’ description of the guidelines makes it sound like a beta-lactam alone or a beta-lactam plus a respiratory quinolone are recommended options for non-severe inpatient CAP, which is not the case.

We are sincerely sorry; it was a mistake and will reformulate the sentence:

“both European guidelines (EG) and American guidelines (AG) recommend empiric therapy with a non–anti-Pseudomonal beta-lactam, plus or minus a macrolide or respiratory fluoroquinolone” is changed to “both European guidelines (EG) and American guidelines (AG) recommend empiric therapy with a non–anti-Pseudomonal beta-lactam and a macrolide or a quinolone alone in non-severe inpatient pneumonia, while recommend a non-anti-Pseudomonal beta-lactam plus either a macrolide or a quinolone in severe inpatient pneumonia for patients without risk factors for PA or MRSA”

  • Instead of referring to the ATS/IDSA guidelines as “American guidelines” I think it is more appropriate to refer to the guidelines by the organizations ATS/IDSA that wrote the guidelines.

According to your indication “American guidelines” is changed to “ATD/IDSA” in all the sentences sections

  • Line 292 – I think it is more accurate to state that if there is prior respiratory isolation of Pseudomonas, then the ATS/IDSA recommend pseudomonal coverage. The phrase “prior infection” used by the authors is a little vague.

In order to be more specific “such as prior infection, colonization” is changed to “such as prior respiratory isolation of Pseudomonas”

  • Line 293 – 294 – I don’t see where the ATS/IDSA guidelines recommend using double pseudomonal coverage for patients with PA risk factors. In Table 4 of the guidelines, the ATS/IDSA CAP guidelines reference the 2016 HAP/VAP guidelines and recommend anti-PA beta-lactams alone. I don’t believe the 2019 ATS/IDSA CAP guidelines ever mention aminoglycosides in any capacity.

We completely agree with your observation, although supported by a recent editorial (Torres A, Niederman MS. Too Much or Too Little Empiric Treatment for Pseudomonas aeruginosa in Community-acquired Pneumonia? Ann Am Thorac Soc. 2021 Sep;18(9):1456-1458), combination therapy is not supported by the evidence coming from most clinical studies and we will delete this sentence according to your suggestion.

  • It is probably worth mentioning that the 2016 IDSA HAP/VAP guidelines eliminated the term healthcare-associated pneumonia in favor of the dichotomy of CAP or HAP, which has led to less broad spectrum antimicrobial use and presumably less use of anti-PA agents for empiric CAP treatment (which may have previously been considered HCAP instead of CAP).

 We agree this choice was justified by the fact that increasing spectrum of therapy by epidemiological criteria alone (HCAP vs CAP) would lead to over-treatment without impact on outcome. To improve prescriptive appropriateness LGs suggest focusing more on individual risk factors and severity scores than on patient provenance.

We will add a sentence about it (line 302-306):

“In 2016 ATS/IDSA guidelines for hospital-acquired pneumonia (HAP) and ventilator-associated pneumonia (VAP), the term healthcare-associated pneumonia (HCAP) was eliminated in favor of the dichotomy of CAP or HAP, leading to less broad spectrum antimicrobial use and anti-PA agents for empiric treatment for CAP previously considered HCAP.”

As I mentioned earlier, I recommend that someone proofread the entire manuscript for grammatical mistakes. An example of some edits I recommend are:

  • Table 2 – the patient in the Huhulescu et al. study in Austria was presumably exposed to hot tub [not tube] water
  • Table 2 – It looks like the E in ceftriaxone was cutoff
  • Lines 179 – 181 – I think the sentence need to be modified grammatically to something like “Most cases started with empiric antibiotic therapy that did not perfectly cover Pa and were subsequently modified..”
  • Lines 183 – 185 – Among survivors, 66% (8/12) [were] discharged after a [median] recovery of 23.75 days (7 – 35 days); the [median] recovery of patients…
  • Line 309 – why is community acquired pneumonia written out if the acronym CAP is used throughout the rest of the manuscript?
  • Line 316 – I do not believe a comma is needed after “PES score”
  • Line 320 – “…could be responsible [for] CAP with a…”
  • Line 322 – “…coinfection with [a respiratory] virus”

Thanks for your recommendation: according to our suggestion, after a grammatical revision, we correct the following mistakes:

Line 104: “procalcitonine” is changed to “procalcitonin”

Line 105: “troponine” is changed to “troponin”

table 1: “tube” is changed to “tub”

table 1: “Ceftriaxone” is changed to “ceftriaxone”

Line 186-187: “Most cases started with empirical antibiotic, not perfectly covering PA, subsequently changed accordingly with the identification of the pathogen” is changed to “Most cases started with empirical antibiotic therapy that I perfectly cover PA and were subsequently modified accordingly with the identification of the pathogen”

Line 191: “Among survivors, 66% (8/12) was discharged after a medium recovery of 23,75 days (7-35 days); the medium recovery of patients” is changed to “Among survivors, 66% (8/12) were discharged after a median recovery of 23,75 days (7-35 days); the median recovery of patients”

Line 199: “are reported as risk” is changed to “were reported as risk”

“community acquired pneumonia” is changed to “CAP” in all the manuscript

Line 371: “could be responsible of” is changed to “could be responsible for”

Line 372: “…coinfection with other respiratory virus” is changed to “…co-infection with a respiratory virus”

Line 370: “PA-CA” is changed to “PA-CAP”

Thanks for your suggestions,

Best regards,

M. Meschiari

N. Barp

Reviewer 3 Report

A very interesting manuscript combining a case report of a patient with a fatal course of CAP caused by Pseudomonas aeruginosa strain and a mini-review of Pseudomonas aeruginosa CAP is presented.

The manuscript is very well prepared, the clinical course of the patient is well described, and I also highly appreciate the analysis of virulence factors of isolated Pseudomonas aeruginosa strain. I am convinced that this is a very actual topic that clearly fits into the issue of bacterial infections and AMR.

I have the following minor comments:

1) In the text, the wrong names of bacteria are used (Micoplasma pneumoniae - the correct one is Mycoplasma pneumoniae, Chlamydia pneumoniae - the correct one is Chlamydophila pneumoniae).

2) The term non-anti-Pseudomonal b-lactam is somewhat unusual. I recommend considering, for example, a non antipseudomonal betalactam antibiotic.

3) The term Minimal inhibitory Concentration should be given as Minimal inhibitory concentration.

4) The authors report that the isolated strain of Pseudomonas aeruginosa had a good sensitivity profile. I do not consider this to be sufficient, and I recommend specifying exactly the susceptibilities to antibiotics, especially for applied cefepime.

Author Response

Dear reviewer 3,

We are very grateful for your constructive comments and suggestions to our paper entitled: A fatal case of Pseudomonas aeruginosa community-acquired pneumonia in an immunocompetent patient: clinical and molecular characterization and literature review”

We here provide a point-by-point reply to the comments, and we have incorporated the related changes in the manuscript.

We thank you for your thoughtful insights which helped to significantly improve the manuscript.

- “A very interesting manuscript combining a case report of a patient with a fatal course of CAP caused by Pseudomonas aeruginosa strain and a mini-review of Pseudomonas aeruginosa CAP is presented.

The manuscript is very well prepared, the clinical course of the patient is well described, and I also highly appreciate the analysis of virulence factors of isolated Pseudomonas aeruginosa strain. I am convinced that this is a very actual topic that clearly fits into the issue of bacterial infections and AMR.

I have the following minor comments:”

1) In the text, the wrong names of bacteria are used (Micoplasma pneumoniae - the correct one is Mycoplasma pneumoniaeChlamydia pneumoniae - the correct one is Chlamydophila pneumoniae).

Thanks for your suggestion: we correct the wrong term as you suggested.

2) The term non-anti-Pseudomonal b-lactam is somewhat unusual. I recommend considering, for example, a non antipseudomonal betalactam antibiotic.

According to your suggestion, “b-lactam” is changed to “non anti-pseudomonal beta-lactam”

3) The term Minimal inhibitory Concentration should be given as Minimal inhibitory concentration.

The term is changed according to your suggestion

4) The authors report that the isolated strain of Pseudomonas aeruginosa had a good sensitivity profile. I do not consider this to be sufficient, and I recommend specifying exactly the susceptibilities to antibiotics, especially for applied cefepime.

Thanks for your suggestion; we will add a sentence about it (line 139-143) :

“Microbiological investigations revealed that blood and bronchoalveolar lavage culture were positive for PA with a good sensitivity profile (susceptible to meropenem MIC 1ưg/ml, amikacin MIC 4ưg/ml, ceftazidime/avibactam MIC 2ưg/ml and ceftolozane/tazobactam MIC 0.5ưg/ml, intermediate to piperacillin/tazobactam MIC 8ưg/ml, ceftazidime MIC 2ưg/ml, cefepime MIC 2ưg/ml and ciprofloxacine MIC 0.25ưg/ml).”

Thanks for your suggestion,

Best regards,

M. Meschiari

N.Barp